# Evaluating Serum Calcium and Magnesium Levels as Predictive Biomarkers for Tuberculosis and COVID-19 Severity: A Romanian Prospective Study

**DOI:** 10.3390/ijms25010418

**Published:** 2023-12-28

**Authors:** Ramona Cioboata, Corina Maria Vasile, Mara Amalia Bălteanu, Dragos Eugen Georgescu, Claudia Toma, Amelia Sanda Dracea, Dragos Nicolosu

**Affiliations:** 1Pneumology Department, University of Medicine and Pharmacy, 200349 Craiova, Romania; ramona.cioboata@umfcv.ro; 2Pneumology Department, Victor Babes University Hospital Craiova, 200515 Craiova, Romania; nicolosud@yahoo.com; 3Department of Pediatric and Adult Congenital Cardiology, University Hospital of Bordeaux, F-33600 Bordeaux, France; corina.vasile93@gmail.com; 4Department of Pneumology, “Marius Nasta” Institute of Pneumology, 050159 Bucharest, Romania; 5Department of Pulmonology, Faculty of Medicine, Titu Maiorescu University, 031593 Bucharest, Romania; 6“Carol Davila” Faculty of Medicine, University of Medicine and Pharmacy, 050474 Bucharest, Romania; gfdragos@yahoo.com; 7Department of General Surgery, “Dr. Ion Cantacuzino” Clinical Hospital, 022904 Bucharest, Romania; 8Pneumology Department, University of Medicine Carol Davila, 020021 Bucharest, Romania; claudiatoma@yahoo.co.uk; 9Department of Biophysics, University of Medicine and Pharmacy of Craiova, 200349 Craiova, Romania

**Keywords:** tuberculosis, COVID-19, serum calcium, serum magnesium, disease severity, biomarkers

## Abstract

In Romania, the highest incidence of tuberculosis (TB) within the European Union was reported in 2020, highlighting a significant health challenge. This is compounded by the COVID-19 pandemic, which has severely impacted healthcare services, including TB management. Both TB and COVID-19, diseases with considerable morbidity and mortality, have shown potential links to electrolyte imbalances. We conducted a prospective study at Victor Babes Hospital, Romania on 146 patients (74 with TB, 72 with COVID-19) between December 2021 and July 2023. This study assessed correlations between disease severity and serum calcium and magnesium levels, as well as pulmonary function. Adult patients with confirmed diagnoses and comprehensive medical records were included, excluding those with chronic respiratory diseases or unrelated electrolyte imbalances. Statistical analysis utilized the Kruskal–Wallis test and Dunn’s procedure for non-normally distributed data. Low serum calcium and magnesium levels were significantly correlated with severe forms of TB and COVID-19, suggesting their potential as biomarkers of disease progression. Patients with more severe TB (i.e., multiple cavities) exhibited significantly lower serum calcium (*p* = 0.0049) and magnesium levels (*p* = 0.0004). ROC analysis revealed high AUC values for serum calcium and serum magnesium in predicting COVID-19 severity, indicating their potential as biomarkers. This study demonstrates a significant association between lower serum calcium and magnesium levels and increased TB severity. Similarly, these electrolytes show promise as predictive markers for COVID-19 severity. These findings could serve as biomarkers for predicting the severity of TB and COVID-19, offering potential utility in clinical decision-making.

## 1. Introduction

Tuberculosis (TB), caused by *Mycobacterium tuberculosis*, is a major global health concern, predominantly spread via respiratory transmission, and responsible for infecting approximately one-quarter of the world’s population [1]. Despite this high infection rate, only an estimated 5–10% of infected individuals develop TB in their lifetime [2]. TB predominantly affects adults, with a higher incidence in males, and can target multiple organs, most commonly the lungs. Without effective treatment, TB has a high mortality rate: approximately 50% [3]. The burden of TB is geographically skewed, with the majority of cases and deaths concentrated in specific regions of Asia and Africa. Influenced by factors such as undernourishment, diabetes, HIV infection, alcohol use disorders, and smoking, TB was the leading cause of death from a single infectious agent, surpassing HIV/AIDS, until the advent of COVID-19 [4,5].

In 2021, there was a notable increase in the global incidence of tuberculosis (TB), with 10.6 million cases reported—a 4.5% rise from 2020, ending a long-term trend of decline. This surge was most pronounced in Southeast Asia (45%), Africa (23%), and the Western Pacific (18%), with adult men being the most affected group, comprising 57% of the cases. The COVID-19 pandemic complicated incidence estimations, necessitating reliance on dynamic models due to disrupted national disease surveillance [3].

Simultaneously, TB mortality also rose significantly, reversing the declining trend observed since 2005. Approximately 1.6 million deaths were recorded in 2021, up from 1.5 million in 2020, with the majority among HIV-negative individuals. This increase undid progress towards the End TB Strategy’s goal of reducing TB deaths by 35% from 2015 to 2020, with only a 5.9% reduction achieved by 2021. This higher mortality can be attributed to pandemic-related disruptions in TB services, particularly in case detection and treatment, which have a rapid and severe impact on mortality rates [3].

The recent global health crisis, initiated by the highly pathogenic human coronavirus, severe acute respiratory syndrome coronavirus 2 (SARS-CoV-2), has led to the most severe pandemic of the last century, resulting in coronavirus disease 2019 (COVID-19) [6]. Characterized by significant mortality and morbidity, the outbreak began in December 2019 with unusual viral pneumonia cases in Wuhan, Hubei Province, China, and it quickly spread globally [7]. As of mid-2023, the WHO has shifted its approach, recognizing COVID-19 as an established health issue and advising on transitioning to its long-term management [8]. This pandemic has significantly disrupted healthcare services worldwide, impacting access to essential TB services and leading to pronounced drops in TB case notifications [9,10]. The end of 2022 marked the beginning of a recovery in essential health services, as per WHO reports. However, economic recovery, especially in emerging economies and disadvantaged groups, is anticipated to take longer [11,12].

In 2020, Romania reported the highest incidence of TB within the European Union/European Economic Area (EU/EEA), accounting for 23.2% of the 33,148 TB cases reported across 29 EU/EEA countries. This stark contrast underscores Romania’s unique challenge with TB, marking it as the European country with the highest TB incidence rate during that period [13]. Additionally, Romania has recorded a total of 3,501,193 confirmed COVID-19 cases, with 68,590 deaths [14].

Serum magnesium maintains normal cellular function and metabolism, and it is crucial for various enzymatic reactions, ion channel regulation, and energy production. The interaction between SARS-CoV-2 and ACE-2 receptors, found in pneumocytes and endothelial cells, is facilitated by the virus’s S-spike protein. This interaction alters ACE-2 pathways, leading to acute damage in the lungs, heart, and endothelial cells [15].

A serum magnesium deficiency might contribute to the pathophysiology of SARS-CoV-2 infection. During the progression of COVID-19, a lack of serum magnesium can increase the risk of initiating a severe inflammatory response known as a “cytokine storm”, harm the vascular endothelium, and trigger a coagulation cascade that could result in disseminated intravascular coagulation. Magnesium, an essential mineral, plays a pivotal role in the optimal functioning of various immune cells, including T cells, B cells, and macrophages. These cells are integral to the body’s defense mechanisms and crucial for cytokine production. Serum magnesium deficiency can disrupt cellular functions, potentially leading to an exaggerated inflammatory response. This is particularly relevant in inflammatory diseases like COVID-19, where cytokine production is a critical factor.

Increased levels of pro-inflammatory cytokines, such as TNF-α, IL-6, and IL-1β, are linked to serum magnesium deficiency. These cytokines are central to developing a cytokine storm, a severe inflammatory response that can occur in diseases like COVID-19 and lead to dangerous complications [16]. Ashique et al. [16] highlighted serum magnesium’s role in mitigating oxidative stress, closely related to inflammation. In conditions like COVID-19, oxidative stress can intensify the inflammatory response, leading to severe complications. Serum magnesium’s ability to reduce oxidative stress is crucial in modulating inflammation and preventing exacerbated responses, such as cytokine storms. Additionally, serum magnesium functions as a natural antagonist to serum calcium. It affects the activity of NMDA receptors involved in inflammation. A deficiency in serum magnesium may increase serum calcium influx, further stimulating the release of inflammatory cytokines and exacerbating the inflammatory response. Given its critical role in immune function and inflammation regulation, serum magnesium supplementation is suggested as a potential therapeutic approach for reducing the severity of inflammatory diseases, including COVID-19. Maintaining adequate serum magnesium levels controls immune responses and prevents excessive inflammation.

Furthermore, serum magnesium may play a role in the onset of long COVID-19 syndrome, potentially exacerbating symptoms or existing health conditions [17]. A study indicated that 70% of acutely ill patients exhibited reduced serum calcium levels (−0.25 to −0.2 mmol/L), linked with a poorer prognosis [17]. Earlier research on COVID-19 patients revealed that severe or critical cases often involve electrolyte disturbances, including serum calcium level imbalances [18].

In recent medical research, the assessment of serum calcium levels has emerged as a crucial factor in understanding and managing various infectious diseases, including tuberculosis (TB) and COVID-19. Serum calcium, a vital electrolyte in the body, plays a significant role in numerous physiological processes, including cell signaling, muscle function, and vascular contraction. Its levels in the blood can reflect changes in bodily function and disease states.

Our decision to assess serum calcium levels stems from their potential as diagnostic and prognostic biomarkers in infectious diseases. In TB, for example, a study by Chandra TJ [19] on 100 participants highlighted a correlation between serum calcium levels and disease severity, as indicated by smear grading. Although the statistical significance was not strong, the trend observed suggests an inverse relationship between serum calcium levels and TB severity [19].

Similarly, in the context of the ongoing COVID-19 pandemic, understanding the multifaceted pathophysiology of the virus is crucial. A 2023 study involving 127 COVID-19 patients revealed that lower serum calcium levels are common in these patients, regardless of disease severity. Particularly in severe or critical cases, most exhibited low serum calcium levels. This trend, alongside correlations with severe organ injuries and elevated levels of pro-inflammatory cytokines like IL-6, underscores the importance of serum calcium in the pathophysiological mechanisms of COVID-19 [20].

This study aims to thoroughly investigate the relationship between electrolyte imbalances, specifically serum calcium and serum magnesium levels, and the severity of COVID-19 and TB. We intend to analyze whether these electrolyte disturbances correlate with the severity of these diseases. Additionally, our study will explore the potential impact of such imbalances on the pulmonary function parameters, including forced expiratory volume in 1 s (FEV1), forced vital capacity (FVC), maximum expiratory flow at 50% of FVC (MEF 50), and the FEV1/FVC ratio, in patients diagnosed with COVID-19 and TB. This research aims to provide deeper insights into the pathophysiology of these infections and contribute to developing more effective clinical management strategies for patients suffering from these diseases.

## 2. Results

### 2.1. General Results

Our study involved 146 patients (74 with tuberculosis and 72 with COVID-19), aged 27 to 86 years, categorized into four groups: moderate and severe COVID-19, and two TB categories (TB1-9 and TB1+,2+,3+). This classification was based on respiratory symptoms, chest imaging, and systemic involvement. The baseline characteristics of these participants are detailed in Table 1.

Our study assessed paraclinical parameters (D-dimers, serum calcium, magnesium levels, pulmonary function tests, and platelet counts) in the four patient groups. Significant differences among these groups were found using the Kruskal–Wallis test, as detailed in Table 2.

Our study found progressive increases in D-dimers from the TB groups to severe COVID-19, indicating escalated clotting activity. Both serum calcium and serum magnesium levels were lowest in severe COVID-19, aligning with disease severity. Pulmonary tests showed better lung function in moderate COVID-19 at 1 and 6 months. Platelet counts were lower in COVID-19 patients than in TB patients, suggesting an impact on platelets. The Kruskal–Wallis test confirmed significant differences across groups, highlighting the variable clinical and physiological impacts of TB and COVID-19 severity levels.

### 2.2. D-Dimers, Serum Calcium, and Serum Magnesium Levels with Respect to TB and COVID-19 Severity

D-dimer levels, indicative of coagulation activity, showed significant variation among the groups. Patients with severe COVID-19 had the highest levels, pointing to a heightened coagulation response. In contrast, TB patients displayed lower levels, suggesting reduced activity. Statistically, severe COVID-19 patients had much higher D-dimer levels than those with moderate cases (*p* < 0.0001), and both COVID-19 groups had higher levels compared to TB patients. The TB subgroups did not differ significantly (*p* = 0.8504), indicating a uniform coagulation profile within these categories. These trends are visually represented in Figure 1, where the color intensity on the heatmap correlates with the strength of statistical differences between groups—yellow signifies strong significance (*p*-values close to 0), while red indicates non-significance (*p*-values close to 1).

Serum calcium levels, which reflect disease severity, were analyzed across the patient groups. Moderate COVID-19 patients had stable levels, while severe cases showed a decrease, suggesting a severity correlation. TB patients with more extensive disease had similar levels to severe COVID-19 patients, but the TB 1-9 group had notably higher levels. The Kruskal–Wallis test confirmed these differences (*p* < 0.0001), highlighting the potential impacts of disease severity on calcium homeostasis. This is represented in Figure 2, where darker shades in the heatmap indicate smaller statistical differences (*p*-values closer to 1).

Our statistical analysis revealed discernible differences in serum magnesium levels across the study groups. Significant disparities were noted between patients with moderate and severe COVID-19 (*p* < 0.0001), indicating potential severity-related alterations in magnesium metabolism. A similar pattern of significance was observed between both COVID-19 groups and the TB1+,2+,3+ group (*p* < 0.0001 for moderate COVID-19; *p* = 0.0014 for severe COVID-19), suggesting a possible disease-specific impact on serum magnesium. In contrast, the TB 1-9 group did not significantly differ from the moderate COVID-19 group (*p* = 0.1372), implying a consistent magnesium profile in less severe disease presentations. However, a highly significant difference was found when comparing the TB 1-9 group to the severe COVID-19 group (*p* < 0.0001), underscoring a pronounced deviation in magnesium levels corresponding to disease severity, as depicted in the heatmap, with varying shades of blue indicating the strength of the correlations (Figure 3).

### 2.3. Pulmonary Function Tests’ Correlation with the Form of TB and COVID-19

In our study, significant variations in pulmonary function tests were observed across the different patient groups. Moderate COVID-19 patients consistently demonstrated higher FEV1 and FVC values at 1 and 6 months, indicating better lung function and less airway obstruction compared to both TB groups and severe COVID-19 cases. The TB 1-9 group’s results were akin to those of the severe COVID-19 group, while TB1+,2+,3+ patients generally showed lower values. These findings underscore the varied impact of disease severity and type on respiratory function.

### 2.4. Platelets and Hemoglobin

Severe COVID-19 patients had markedly elevated PLT levels compared to those with moderate COVID-19, reflecting a significant difference (*p* < 0.0001). Both the moderate and severe COVID-19 patient groups had higher PLT levels than the TB patients, indicating an escalated coagulative response in the context of the viral infection (*p* < 0.0001 for both when compared to the TB groups). No significant variation in PLT levels was observed between the TB subgroups (*p* = 0.8504), denoting a consistent coagulation profile across these categories. Figure 4 shows a heatmap representing the PLT levels, where the color intensity correlates directly with the statistical significance of the differences between patient groups. A more neutral color intensity indicates significant differences in PLT levels between severe COVID-19 patients and the other groups (*p* < 0.0001). In contrast, a strong color reflects the lack of significant difference in PLT levels within the TB subgroups (*p* = 0.8504). This visual tool effectively communicates the relative magnitude of coagulation activity as indicated by the PLT levels among the patient groups.

There were no significant differences in hemoglobin levels between patients with moderate and severe COVID-19 (*p* = 0.3781), indicating that disease severity is not correlated with hemoglobin variation.

Both COVID-19 groups exhibited significantly lower hemoglobin levels compared to the TB groups, with *p*-values indicating strong statistical significance (*p* < 0.0001 for moderate COVID-19 vs. TB1+,2+,3+; *p* = 0.0007 for moderate COVID-19 vs. TB 1-9; and *p* < 0.0001 for severe COVID-19 comparisons); this can be observed in Figure 5, shown in yellow.

Among the TB patients, those in the TB1+,2+,3+ group had significantly lower hemoglobin levels than those in the TB 1-9 group, but the difference was less significant (*p* = 0.0316).

### 2.5. Analysis of ROC Curve Efficacy in Discriminating Disease Severity for COVID-19 and TB

The ROC curves in Figure 6 assess the effectiveness of certain clinical parameters in classifying the severity of COVID-19 and TB. Parameters with AUC values near 1, such as serum calcium and magnesium, are highly effective in distinguishing between disease severities. Lower AUC values for hemoglobin and platelets suggest these are less effective. Clinical measures with AUCs significantly above 0.5 and high sensitivity and specificity are deemed reliable for predicting patient severity.

In the provided ROC curves, each line represents the diagnostic accuracy of a clinical parameter in distinguishing disease severity. High AUC values indicate better discrimination capability: serum calcium and magnesium showcase AUCs close to 1, denoting high diagnostic accuracy, while hemoglobin and platelets, with their lower AUCs, have comparatively limited discriminative value.

The analysis prioritizes parameters with AUCs significantly above 0.5 and strong sensitivity and specificity (both above 75%) for reliably identifying disease severity in COVID-19 and TB patients.

The ROC curves depicted in Figure 7 illustrate the ability of various biomarkers to differentiate between categories of TB infection. Hemoglobin and platelets show AUCs slightly above 0.5, indicating limited utility in distinguishing TB severity levels. D-dimers, with a moderate AUC of 0.696, suggest some usefulness in identifying more severe TB cases. Serum calcium and magnesium levels have high AUCs, demonstrating strong discriminative potential. However, their clinical applicability is hampered by the low sensitivity of serum calcium and low specificity of serum magnesium at their best thresholds.

Pulmonary function tests, not depicted in this figure but mentioned in the text, have AUCs below 0.5, making them unreliable indicators of TB severity. While serum calcium and magnesium are highlighted as significant, their use in clinical practice requires careful threshold calibration for effective TB severity classification.

### 2.6. Serum Calcium and Magnesium Imbalances in Radiological Findings of TB and COVID-19

Figure 8 and Figure 9 present boxplots that correlate TB severity with serum calcium and magnesium levels.

In Figure 8, it can be seen that lower serum calcium levels are significantly associated with patients who have multiple pulmonary cavities, a more severe form of TB (TB1+,2+,3+), compared to those with consolidation or nodular opacities (*p* = 0.0049).

Figure 4 similarly indicates that serum magnesium levels are significantly lower in patients with multiple cavities than in those with less severe TB radiological patterns (*p* = 0.0004).

These findings suggest a clear link between the severity of TB lung damage and diminished levels of key electrolytes, highlighting the potential need for targeted management of serum calcium and magnesium levels in patients with severe TB.

## 3. Discussion

To the best of our knowledge, this study is among the first to concurrently evaluate the implications of COVID-19 and tuberculosis, offering unique insights into their comparative pathophysiology.

We identified several significant associations in our investigation of the correlations between electrolyte imbalances and disease severity in tuberculosis (TB) and COVID-19 patients. Our study found that patients with TB presenting with multiple cavities, indicative of more severe disease, had notably lower serum calcium levels. This relationship was statistically significant and suggests that serum calcium levels may be inversely correlated with the severity of TB’s radiological findings.

Additionally, we observed a highly significant difference in serum magnesium levels between the same patient groups. Those with more severe TB pathology, as denoted by the presence of multiple cavities, also demonstrated lower levels of serum magnesium, hinting at a potential role of this electrolyte in the pathophysiology of TB.

Our analysis extended to assessing various clinical parameters through ROC curves to distinguish between severe and moderate COVID-19 cases. The results pointed to serum calcium and serum magnesium as highly accurate predictors of disease severity, with AUC values close to 1. These electrolytes, D-dimer levels, and pulmonary function test results emerged as potential biomarkers of disease severity in COVID-19, with the latter offering additional insights into respiratory function impairment associated with the infection.

Overall, our findings emphasize the importance of monitoring serum calcium and serum magnesium levels in patients with TB and COVID-19 and suggest that these electrolytes, along with specific clinical parameters, could serve as significant indicators of disease severity, with potential implications for patient management and treatment strategies.

Based on a retrospective study by Zhou et al. [20], involving 127 patients confirmed with COVID-19, a clear correlation was established between the severity of COVID-19 and low serum calcium levels. Their findings indicate that both mild/moderate and severe/critical cases of COVID-19 exhibited low serum calcium levels in the early stages of the infection. Notably, the severe/critical cases demonstrated significantly lower serum calcium levels than the mild/moderate cases at these early stages. Zhou et al. also observed that low serum calcium levels were associated with severe/critical multiorgan injuries, especially in the mild/moderate patient group. Furthermore, they found a correlation between changes in serum calcium levels and the pro-inflammatory cytokine IL-6 across both mild/moderate and severe/critical cases.

This study highlights the prevalence of low serum calcium as a common abnormality in COVID-19 patients, particularly those in severe or critical condition. This underscores the importance of serum calcium as a potential early-stage biomarker for assessing COVID-19 severity, linking it to multiorgan injuries and elevated levels of pro-inflammatory cytokines like IL-6.

These findings are consistent with those of our study, where patients with more severe forms of COVID-19 exhibited significantly lower serum calcium and serum magnesium levels. Our study expands upon Zhou et al.’s work by suggesting the potential of these electrolytes as biomarkers of disease severity in COVID-19. This broader scope encompasses the implications of electrolyte imbalances in COVID-19, reinforcing the clinical importance of monitoring these levels in patients with this disease.

Our results also support the study conducted by Alemzadeh et al. [21], published in 2021, which systematically reviewed and meta-analyzed data on the impact of serum calcium levels on COVID-19 outcomes. Their investigation included 25 articles, revealing that 59% of COVID-19 patients experienced hypocalcemia. They found significant associations between hypocalcemia and the severity of COVID-19 (*p* = 0.002), increased mortality (odds ratio (OR) = 6.99), extended hospitalization (*p* < 0.001), and a higher likelihood of admission to the intensive care unit (OR = 5.09). Additionally, they observed a direct correlation between low serum calcium levels and elevated D-dimer levels (*p* = 0.02) and reduced lymphocyte counts (*p* = 0.007). Consequently, they concluded that lower serum calcium levels are linked to higher mortality and complications in COVID-19 patients, suggesting that serum calcium should be considered as a prognostic factor for disease severity and included in the initial patient assessments.

Guziejko et al. [22] found no significant differences in PFTs between COVID-19 convalescents and a healthy control group. Still, they highlighted that COVID-19 patients with a persistent cough had decreased lung function parameters. At the same time, our study showed significant differences in FEV1 and FVC values among patients with severe and moderate COVID-19 and different TB stages. Both studies underscore the importance of PFTs in the follow-up of patients who have suffered from COVID-19, albeit with other implications. Our study suggests a need for differentiated follow-up strategies based on the severity of COVID-19 and TB. At the same time, Guziejko et al. recommend PFT for patients with persistent symptoms like cough, regardless of the initial severity of their COVID-19 infection.

Coagulation abnormalities in COVID-19 patients often lead to the progression of the disease towards severe conditions and potentially fatal outcomes, a process marked by increased D-dimer levels and the formation of thrombi in both veins and arteries—elevated D-dimer in COVID-19 results from excessive clotting and low oxygen levels in the blood. Additionally, an increase in D-dimer levels is commonly seen in patients with severe COVID-19 and is significantly linked to higher mortality rates. As D-dimer is a breakdown product of fibrin, its presence serves as an indicator for the likelihood of pulmonary embolism and deep venous thrombosis (DVT) [23,24,25].

The D-dimer levels observed in our study of COVID-19 patients were consistent with these established findings. We observed elevated D-dimer levels, particularly in patients with severe COVID-19, which is consistent with the model that higher D-dimer levels are associated with more severe disease and higher mortality risk. Consistent with its role as a fibrin degradation product, the elevated D-dimer levels in our cohort also suggest an increased risk of thrombotic complications such as pulmonary embolism and deep vein thrombosis, reinforcing its utility as a critical biomarker in the management of COVID-19.

The study by Baquri and Kora [26] investigating the relationship between serum magnesium levels and pulmonary tuberculosis severity showed significant results. They observed a marked decrease in serum magnesium levels in patients with TB, which became more pronounced in the advanced stages of the disease. Their research concluded that serum magnesium levels were inversely correlated with TB severity, indicating its potential as a valuable biomarker for assessing disease progression. This inverse relationship suggests that the need for serum magnesium supplementation may increase as TB progresses, highlighting a potential area for therapeutic intervention.

In comparison, our study investigated serum magnesium levels in different stages of TB, specifically in the TB 1-9 and TB1+,2+,3+ categories. Our findings revealed that while serum magnesium levels were lower in all TB patients, serum magnesium deficiency had a notable gradation correlating with the severity of TB. Patients in the TB1+,2+,3+ categories, representing more advanced stages of TB, exhibited more significant decreases in serum magnesium levels than those in the TB 1-9 category. This gradation further supports the hypothesis that serum magnesium levels could serve as an indicator of disease severity in TB patients. Our results are consistent with those of Baquri and Kora’s study, reinforcing the concept of serum magnesium levels as a potential biomarker of TB severity and suggesting the need for serum magnesium supplementation as part of TB treatment protocols, especially in more advanced cases.

The role of serum magnesium in the immune response, particularly its potential in mitigating disease severity, aligns with the observed decrease in serum magnesium levels in more advanced stages of TB in this study. This parallel suggests that serum magnesium deficiency might be a common thread in the progression of various infectious diseases, including TB and COVID-19.

Moreover, the potential therapeutic applications of serum magnesium, as indicated in Guerrero-Romero et al.’s study on COVID-19, could be relevant for TB. Considering the decrease in serum magnesium levels in advanced TB, exploring serum magnesium supplementation might be beneficial in managing TB, similar to its proposed role in COVID-19.

In summary, Guerrero-Romero et al.’s [27] study underscores the importance of serum magnesium in infectious diseases. It provides a valuable reference point for discussing our findings on serum magnesium levels in TB patients. It highlights the potential of serum magnesium not only as a biomarker for disease severity, but also as a possible therapeutic avenue in managing diseases like TB and COVID-19.

As stated in recent research [28], it is essential to underline the critical importance of addressing the challenges posed by COVID-19 and TB management, mainly focusing on prevention.

In addition to our findings, it is pertinent to consider the insights from a study published last year [29]. This study underscores the long-term health effects of COVID-19 and the significance of conducting extended follow-up studies. Similar to our observations, the study revealed a range of persistent symptoms in COVID-19 patients post-discharge, including fatigue, cough, and myalgia. Notably, it was reported that many patients experienced various post-discharge symptoms, with durations extending between 19 and 26 days. The study also pointed out that patients with severe forms of COVID-19 exhibited more severe lung diffusion impairment and abnormalities in lung imaging. This is consistent with our findings, where severe COVID-19 cases demonstrated lower serum calcium and serum magnesium levels, pointing towards an intricate relationship between disease severity and electrolyte imbalances.

Furthermore, the study’s findings on post-COVID-19 syndrome (long COVID-19) as a multisystem disease with a complex long-term impact resonate with our observations. These insights emphasize the need for comprehensive recovery programs and multidisciplinary collaboration to manage post-COVID-19 patient health effectively. This holistic approach is crucial for addressing the multifaceted challenges posed by both COVID-19 and TB, underscoring the need for vigilant monitoring and tailored treatment strategies for patients afflicted with these diseases.

In the context of our study’s findings, it is crucial to consider the complexities of diagnosing COVID-19, as highlighted by Marginean et al. [30]. Given the pandemic’s urgency, this study stresses the importance of prompt recognition, isolation, and rapid treatment in cases of suspected COVID-19. It underscores the risks of delayed diagnosis and treatment due to the similarity of COVID-19 symptoms and imaging to those of other systemic conditions. Notably, this study points out that one in five patients with respiratory symptoms may be diagnosed with an alternative condition, underlining the complexity and potential for misdiagnosis in the clinical setting. This is particularly significant in our study, which examined electrolyte imbalances in tuberculosis and COVID-19 patients, as it emphasizes the need for thorough clinical and radiological evaluation to avoid misdiagnosis. The analysis further elaborates on the progression of COVID-19 pneumonia to ARDS, characterized by patchy ground-glass confluent areas and pulmonary condensations on HRCT. These insights are particularly relevant to our study’s findings, as they provide a deeper understanding of the diagnostic challenges and the critical need for accurate differentiation between TB, COVID-19, and other respiratory conditions.

A study by Udristoiu et al. [31] explored the association of severe COVID-19 with various clinical parameters and comorbidities. It was found that severe COVID-19 was significantly correlated with several factors: CXR severity; respiratory function parameters like oxygen saturation and respiratory rate; cardiovascular function indicators such as systolic and diastolic blood pressure and cardiac frequency; associated diseases, including diabetes, cardiac and kidney disease, hypertension, autoimmune thyroiditis, and obesity; and the presence of symptoms like coughing, headache, shortness of breath, and others. Moreover, hematological analysis showed a decrease in white blood cells and lymphocytes in cases of severe disease. These findings provide a comprehensive view of the multifactorial nature of severe COVID-19 and underline the complexity of managing patients with such diverse clinical presentations. This resonates with our study’s findings on the electrolyte imbalances in tuberculosis and COVID-19 patients, further emphasizing the intricate interplay of various clinical factors in these diseases.

While speculative, it is plausible to consider that the COVID-19 pandemic may have affected the serum magnesium status of the Romanian population, thereby influencing their susceptibility to tuberculosis. This hypothesis warrants further investigation, considering Romania’s high TB incidence.

## 4. Materials and Methods

### 4.1. Study Design and Population

This prospective study aimed to investigate correlations in disease severity between TB and COVID-19 patients, specifically focusing on serum calcium and serum magnesium levels and various pulmonary function parameters, such as FEV1, FVC, MEF 50, and the FEV1/FVC ratio.

The research was conducted at the Pneumoftiziology Clinic of Victor Babes Hospital in Craiova, Romania, chosen for its extensive patient records and advanced facilities for detailed pulmonary function testing.

Data were collected from December 2021 to July 2023, a period selected to ensure a comprehensive sample size and to capture relevant data during key phases of the COVID-19 pandemic. The study included 146 patients: 72 diagnosed with COVID-19 and 74 with TB, providing a balanced comparison between these groups.

Our study classified TB patients based on the Ziehl–Neelsen staining method and the quantity of acid-fast bacilli (AFB) observed under light/bright-field microscopy. This method categorizes TB infections by counting the number of AFB per microscopic field [32]. The classifications are as follows:

TB 1-9 (0): negative for TB. This is used when no AFB are observed across approximately 300 visual fields spanning two lengths of the slide. It is reported as “0”.

TB scanty positive (1-9): low positive. When 1 to 9 AFB are found within a single length, the count is noted with a “+” sign before the number (e.g., +1, +2, up to +9). This indicates a low but detectable level of TB bacilli.

TB1+,2+,3+ (ranging from 1+ to 3+):

TB1+ (10–99 AFB in 1 length): mildly positive. This is reported as “1+” when 10 to 99 AFB are found in a single length, indicating a mildly positive result for TB.

TB2+ (1–10 AFB per field in at least 50 fields): moderately positive. This category reports 1 to 10 AFB per field in at least 50 visual fields as “2+”. This suggests a moderate level of TB infection.

TB3+ (more than 10 AFB per field in at least 20 fields): highly positive. The “3+” category is assigned when there are more than 10 AFB per field in at least 20 fields, indicating a highly infectious stage of TB.

Radiological findings were also incorporated to complement the TB classification. Multiple cavities observed in radiology indicated TB 1-9, while consolidations or nodular opacities were associated with the TB 1+, 2+, and 3+ categories, providing a more comprehensive understanding of TB severity.

Regarding the severity of illness in adults with SARS-CoV-2, we used the Treatment Guidelines in Coronavirus Disease 2019 [33].

Asymptomatic or pre-symptomatic infection: Patients who test positive for SARS-CoV-2 but without symptoms related to COVID-19.

Mild illness: Patients present signs and symptoms like fever, cough, and headache but do not experience shortness of breath, dyspnea, or abnormal chest imaging.

Moderate illness: Usually characterized by clinical or radiographic evidence of lower respiratory tract disease, with a measured oxygen saturation (SpO_2_) ≥ 94% in ambient air at sea level.

Severe illness: Patients experiencing one or more of the following: oxygen saturation below 94% in room air, PaO_2_/FiO_2_ ratio below 300 mm Hg, respiratory rate above 30 breaths/min, or lung infiltrates over 50%.

Critical illness: The most severe form includes patients experiencing respiratory failure, septic shock, multiple organ dysfunction, the need for mechanical ventilation, and shock.

### 4.2. Inclusion and Exclusion Criteria

Our inclusion criteria were patients aged 18 and older with confirmed diagnoses of TB or COVID-19; as per standard diagnostic procedures for TB, this typically includes a positive sputum test for *Mycobacterium tuberculosis* and chest radiography; for COVID-19, this includes a positive RT-PCR test or rapid antigen test confirming SARS-CoV-2 infection.

Patients were required to have comprehensive medical records, including detailed laboratory results for serum calcium and serum magnesium levels and complete pulmonary function test data. The exclusion criteria were patients with chronic respiratory diseases such as COPD or asthma, which could independently affect the pulmonary function test results, and those with known electrolyte imbalances from causes other than TB or COVID-19. Patients with incomplete medical records or missing essential study data, along with those undergoing treatments that significantly affect serum calcium or serum magnesium levels, were also excluded.

### 4.3. Data Collection and Parameters

Data were collected from the patients’ medical records at the Pneumoftiziology Clinic. These included laboratory reports and pulmonary function test results. The parameters assessed were as follows:

D-dimers: Quantified to assess clot formation and breakdown, providing insights into coagulation abnormalities in TB and COVID-19.

Serum calcium levels: Measured due to serum calcium’s critical role in cellular signaling and structural functions; abnormal levels may indicate underlying pathophysiological processes.

Serum magnesium levels: Assessed for their importance in biochemical reactions and overall bodily functions.

Pulmonary function tests: Included FEV1, FVC, MEF 50, and the FEV1/FVC ratio, providing comprehensive insights into respiratory impact.

### 4.4. Follow-Up

Follow-up evaluations at 1 and 6 months post-diagnosis were conducted to track patient progression and changes in health status. These evaluations included repeat pulmonary function tests, allowing for comparative analysis over these periods.

### 4.5. Statistical Analysis

Statistical analysis was performed using Microsoft Excel (Microsoft Corp., Redmond, WA, USA) and the XLSTAT add-on for MS Excel (Addinsoft SARL, Paris, France). Data were recorded using Microsoft Excel files and then statistically analyzed to find the relationships between the variables of interest for the two groups of patients.

We used the Anderson–Darling and Shapiro–Wilk tests to test the normality of the data. Because the study involved numerical comparisons between groups of patients that did not have a normal (i.e., Gaussian) distribution, the nonparametric Kruskal–Wallis test was primarily used, instead of the ANOVA test, to detect significant differences between the values in the compared data series. If statistically significant differences were detected between the analyzed groups, the Dunn procedure for pairwise comparisons was used to discover the groups with significant differences.

## 5. Conclusions

### Limitations

While our study provides valuable insights into the relationship between electrolyte imbalances and the severity of TB and COVID-19, several limitations warrant mention.

Firstly, the sample size, although sufficient for statistical analysis, was relatively small, which may limit the generalizability of the results. Larger cohort studies could provide more robust data and validate the clinical utility of our findings.

Another area for improvement is that this study was conducted at a single center, which may introduce selection bias and limit the applicability of the results to other settings or populations. Multicenter studies would help confirm the findings’ relevance across diverse clinical contexts.

Also, this study relied on the accuracy and completeness of medical records, which can vary and may affect the reliability of the data. Standardized prospective data collection would enhance the accuracy of the findings.

Finally, while we measured several clinical parameters, other unmeasured factors, such as nutritional status, medication use, or underlying health conditions, could influence electrolyte levels and disease severity. Future studies should aim to control for these potential confounders to clarify the relationships observed.

Despite these limitations, our study contributes to the understanding of TB and COVID-19 and highlights the potential role of serum calcium and serum magnesium levels as markers of disease severity.

In conclusion, our investigation revealed that lower serum calcium and serum magnesium levels are significantly associated with more severe TB radiological findings, with *p*-values of 0.0049 and 0.0004, respectively. The ROC analysis further identified serum calcium and serum magnesium as potential biomarkers for assessing disease severity in COVID-19, with high AUC values suggesting their strong predictive capacity. These results highlight the importance of monitoring these electrolytes in patients with TB and COVID-19.

Future studies with larger sample sizes and across multiple centers are recommended to confirm these findings and to refine the clinical thresholds for these biomarkers, enhancing their utility in patient management.

## Figures and Tables

**Figure 1 ijms-25-00418-f001:**
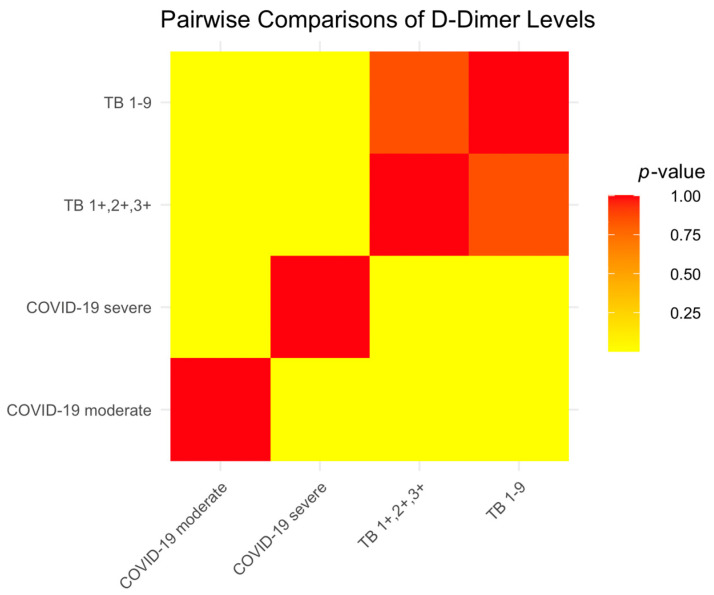
Heatmap of D-dimer level differences by disease severity and type.

**Figure 2 ijms-25-00418-f002:**
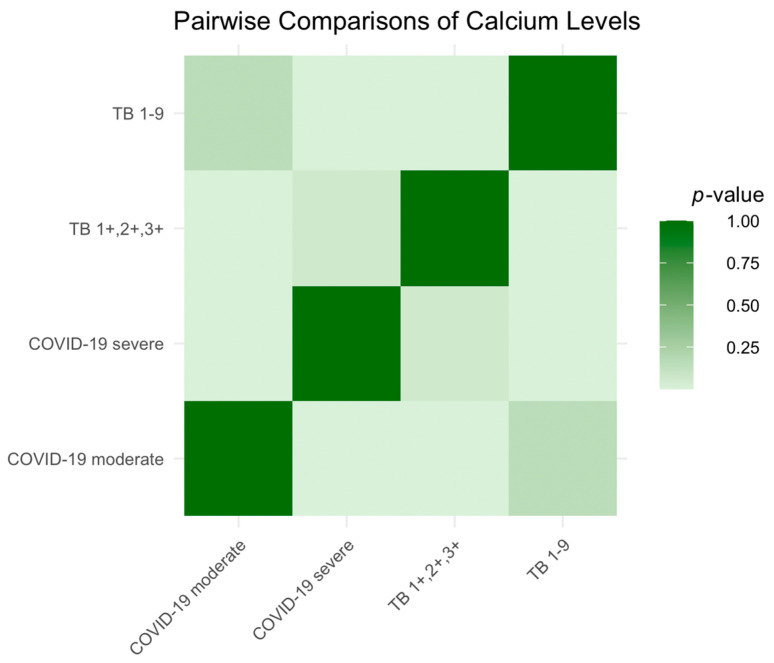
Heatmap of serum calcium level variations across the patient groups.

**Figure 3 ijms-25-00418-f003:**
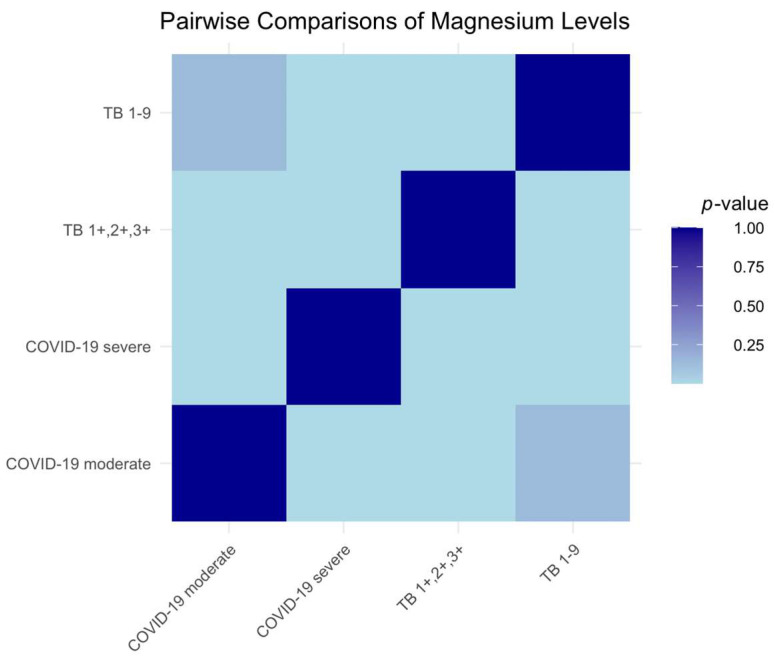
Heatmap of serum magnesium level differences across the patient groups.

**Figure 4 ijms-25-00418-f004:**
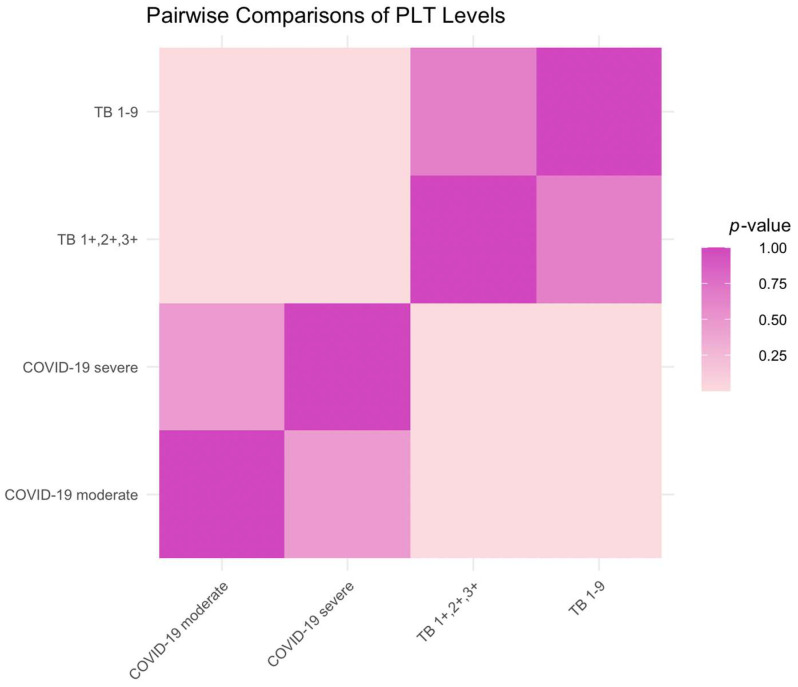
Heatmap analysis of PLT level variations among the patient groups.

**Figure 5 ijms-25-00418-f005:**
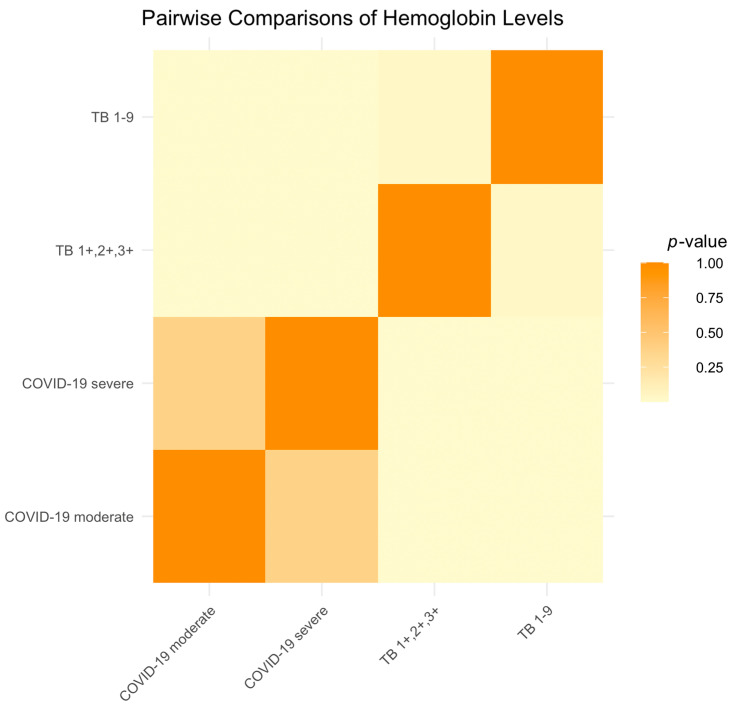
Hemoglobin level significance heatmap across the patient groups.

**Figure 6 ijms-25-00418-f006:**
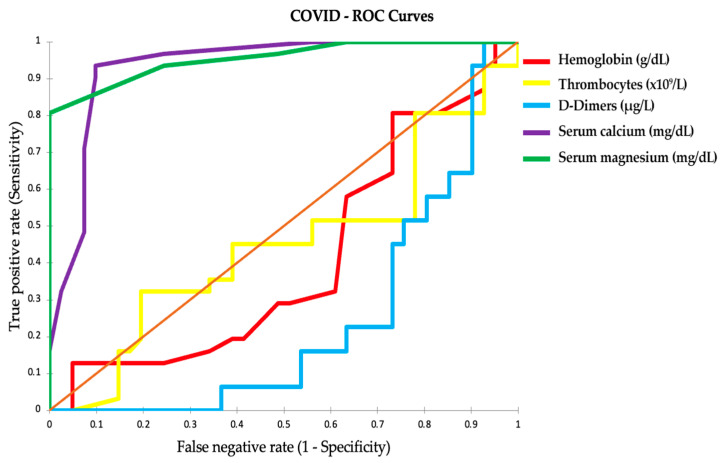
Diagnostic parameter efficacy in COVID-19 and TB severity discrimination: ROC curve analysis.

**Figure 7 ijms-25-00418-f007:**
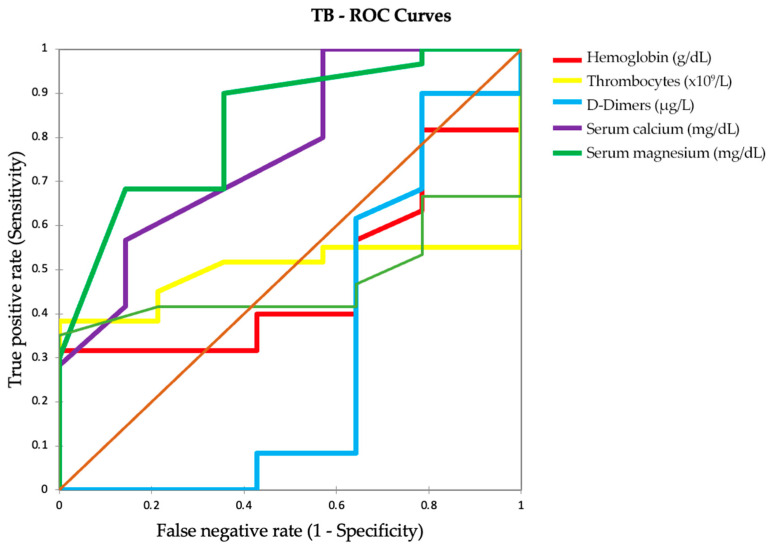
Parameter efficiency in tuberculosis severity discrimination: ROC curve evaluation.

**Figure 8 ijms-25-00418-f008:**
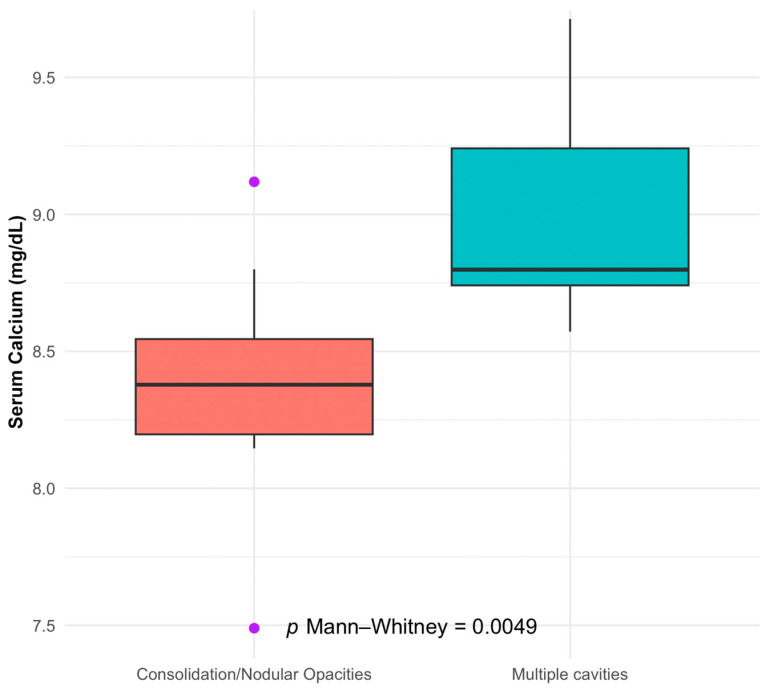
Serum calcium levels in TB’s radiological severity. Purple dots: represent statistical outliers.

**Figure 9 ijms-25-00418-f009:**
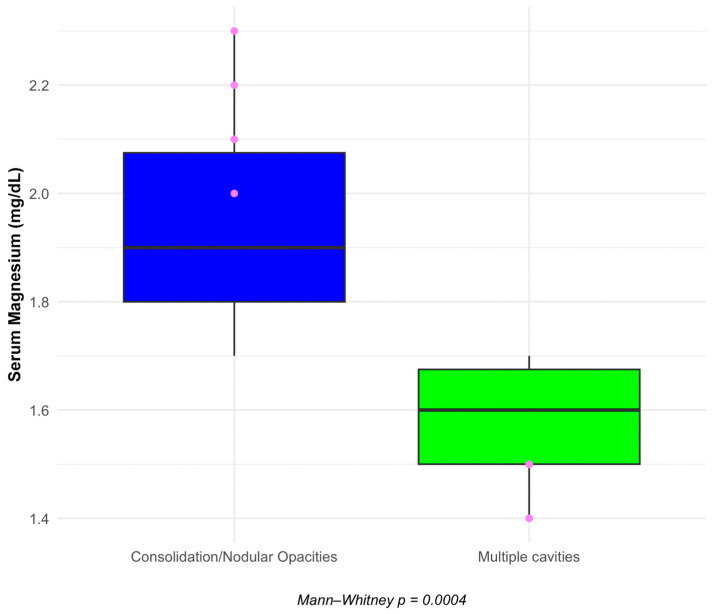
Serum magnesium levels correlated with TB’s radiological severity. Pink dots: represent statistical outliers.

**Table 1 ijms-25-00418-t001:** Baseline characteristics of the study participants.

Variable	COVID-19 N = 72	TB N = 74
**Demographic, Clinical, and Laboratory Characteristics**		
Age (years) [mean values; min–max]	60.74 [27–86]	53.35 [36–67]
Respiratory rate (breaths/min) [average]	29.40	17.32
Cardiac frequency (beats/min) [average]	92.85	89.08
Body temperature (°C) [average]	37.73	36.86
Oxygen saturation (%) [average]	88.94	95.77
PLT (thrombocytes × 10^9^/L) [average]	207,216.67	449,567.5676
Hb (hemoglobin g/dL)	13.96	12.32
WBC (white blood cells × 10^9^/L)	4665.80	8939.77
D-dimers (µg/L)	1419.26 [159.22–5300]	536.29 [226–1053]
**Symptoms and Comorbidities**		
Gender (%female)	52.39	26.49
Smoking (%yes)	52.78	72.97
Coughing (%yes)	77.78	100
Sputum production (%yes)	37.50	100
Shortness of breath (%yes)	61.11	13.51
Palpitations (%yes)	19.44	1.5
Physical astenica (%yes)	79.17	83.78
Abdominal pain (%yes)	25	2
Myalgia (%yes)	44.44	3
Inappetence (%yes)	58.33	85.14
Diabetes (%yes)	29.17	17.57
Cardiovascular disease (%yes)	48.61	21.62
Kidney disease (%yes)	5.56	0
Hypertension (%yes)	45.83	21.62
Obesity (%yes)	41.67	10
**Forms of Disease**		
COVID-19 severe form (%)	43.06	
COVID-19 moderate form (%)	56.94	
TB 1+ to 3+ (%)		63.52
TB 1-9 (%)		36.48

**Table 2 ijms-25-00418-t002:** Comparative analysis of parameters based on the severity of TB and COVID-19.

Parameter	COVID-19 Moderate (41 Cases)	COVID-19 Severe (31 Cases)	TB1+,2+,3+ (47 Cases)	TB 1-9 (27 Cases)	*p*-Value (Kruskal–Wallis)
Coagulation markers					
D-dimers (µg/L)	1147.77 ± 1113.37	2778.32 ± 951.34	558.43 ± 227.80	441.43 ± 289.04	<0.0001
(median, IQR)	739 (508–1113.38)	2567 (1050–3391)	532 (393–567)	567 (320–703)	
Electrolytes					
Serum calcium (mg/dL)	8.07 ± 0.33	7.82 ± 0.25	7.81 ± 0.32	8.36 ± 0.58	<0.0001
(median, IQR)	7.8 (7.1–8.3)	8.1 (7.9–8.2)	7.9 (7.1–8.4)	8.2 (7.5–9.3)	
Serum magnesium (mg/dL)	1.86 ± 0.30	1.31 ± 0.19	1.52 ± 0.20	1.82 ± 0.28	<0.0001
(median, IQR)	1.8 (1.7–2)	1.2 (1.2–1.4)	1.6 (1.4–1.6)	1.9 (1.8–1.9)	
Pulmonary function tests					
FEV1 1 month (% predicted)	77.78 ± 4.85	70.65 ± 6.81	74.27 ± 9.61	73.21 ± 10.81	0.0002
(median, IQR)	78 (76–81)	69 (69–76)	76 (70–81)	72 (62–79)	
FEV1 6 months (% predicted)	81.66 ± 4.84	75.00 ± 6.71	76.03 ± 9.24	73.14 ± 8.87	0.0002
(median, IQR)	83 (79–84)	72 (72–82)	80 (70–83)	70 (62–82)	
FVC1 1 month (% predicted)	75.17 ± 3.64	66.35 ± 2.40	69.57 ± 6.89	66.64 ± 11.09	<0.0001
(median, IQR)	76 (75–77)	65 (65–67)	72 (64–75.5)	65 (61–72)	
FVC1 6 months (% predicted)	79.90 ± 3.85	67.26 ± 13.32	68.28 ± 6.33	75.21 ± 9.76	<0.0001
(median, IQR)	80 (79–81)	68 (68–74)	70 (60–73.5)	75 (73–79)	
MEF 50 1 month (% predicted)	64.15 ± 2.38	60.55 ± 5.21	54.57 ± 13.50	55.64 ± 8.22	0.0001
(median, IQR)	65 (64–65)	62 (59–62)	61 (45–65)	56 (42–68)	
MEF 50 6 months (% predicted)	66.61 ± 3.32	64.87 ± 6.05	57.42 ± 12.63	60.00 ± 4.95	0.0005
(median, IQR)	67 (66–68)	68 (63.5–68)	63 (51–67)	59 (53–70)	
FEV1/FVC 1 month	70.56 ± 1.82	67.65 ± 4.20	66.47 ± 5.78	68.71 ± 3.07	0.0010
(median, IQR)	71 (70–71)	67 (67–70)	69 (64–71)	69 (64–71)	
FEV1/FVC 6 months	71.05 ± 1.97	68.29 ± 4.47	67.28 ± 5.41	70.07 ± 2.27	0.0006
(median, IQR)	72 (70–72)	69 (67–71)	69 (64–72)	71 (66–74)	
Hematological parameters					
Platelets (×10^9^/L)	195,707.32 ± 79,652.76	142,438.71 ± 107,874.02	556,800.00 ± 141,583.04	461,428.57 ± 27,809.00	<0.0001
(median, IQR)	189,000 (139,000–241,000)	145,000 (89,500–192,000)	452,000 (252,000–623,000)	489,000 (420,000–527,000)	
Hemoglobin (g/dL)	13.89 ± 1.35	14.05 ± 1.31	10.35 ± 1.51	12.16 ± 1.22	<0.0001
(median, IQR)	13.9 (13.1–14.9)	14.2 (13.65–14.7)	10.4 (8.975–11.1)	11.85 (11.1–12.9)	

The parameters in our study are presented as means ± standard deviations and medians with IQRs, highlighting both central tendencies and variability within groups. The inclusion of *p*-values from the Kruskal–Wallis test indicates the statistical significance of the observed differences, with *p*-values below 0.05 signifying significant variations among the patient groups.

## Data Availability

More data are available upon request from the corresponding authors.

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
