# Peer review of "Evaluating Serum Calcium and Magnesium Levels as Predictive Biomarkers for Tuberculosis and COVID-19 Severity: A Romanian Prospective Study"

_ijms, 2023, doi:10.3390/ijms25010418_

Round 1

Reviewer 1 Report

Comments and Suggestions for Authors

Cioboata et al., Nutrients submission Review  12-1-23

Electrolyte Imbalances as Indicators of Severity in Tuberculosis and COVID-19 Patients.  A Romanian Prospective Study

Comments to Authors:  You’ve done good work to collect the serum Mg and Ca values on both COVID-19 and TB patients, and your analysis has shown that both these serum electrolyte values are low in these patients, and your ROC AUC analysis predicts that these measures could be biomarkers for progression of these two diseases.

Your presentation is difficult to understand, however, and needs editing and revision to present to your readers a good manuscript that adequately presents your results.

Here are my suggestions:

Title:  I would specific Ca and Mg rather than “electrolytes”.  You don’t really deal with sodium, potassium, or chloride in this study, so the use of Electrolytes in your title is misleading.  You can certainly use “electrolytes” or “serum electrolyes” as a keyword, but I sugget your use the terms “Calcium” and “Magnesium” in your title.

Line 40:  is your citation of ref (1) missing here?

Line 46:  needs a ref for your statement that TB has a high mortality rate, approximately 50%.

Line 73:  needs a reference for this paragraph on ACE-2 path ways.

Line 81:  I suggested this sentence be revised to add as follows:  “A study indicated that 70% of acutely ill patients exhibited reduced serum calcium levels (-0.25 to -0.2 mmol/L) linked with a poorer prognosis (19).

 Section on Materials & Methods:  on line 471 on pg 14 in your Discussion, you refer to “compared to healthy controls” while your materials & methods do not describe any healthy controls.  In addition, your tables of data do not include data on healthy controls.   Did you have a set of healthy controls?  If not, do you mean at line 471 that you are comparing your results with healthy std ref ranges.  Your Methods needs to describe this.  Plus you need to include your healthy control group dataor std healthy ref ranges you used for comparison in your tables.

Results:  Table 1.  I don’t see any units for PLT, Hb, WBD, D-Dimers.   These needs to be included in Table 1s baseline characteristics. 

Table 2.  Again, you don’t list the units for D-dimers, Calcium, Magnesium, FEV measures, MEF measures, or PLT or Hb.   Very important that your tables include what units you are using, i.e. mg/dl for serum Mg and serum Ca values, also, for Magnesim and Calcium, they should be “serum Calcium” and “ serum Magnesium”.  You want your readers to be able to look at your tables and figures and to have a good idea ofl what you’ve done and what you’ve found.   Your data and results are good, but your presentation is poor and hard to discern.  Table 2 needs a better caption, therefore.  What are you comparing?  I don’t understand the two-row presentation you are making here.   What does row 3 (Calcium) have to do with row 4 (<0.0001)?   Your caption has to describe this succinctly and meaningfully.  You might think of bar graphs or some other visual mechanism to display your data. 

Tables 3 – 7:  line 254:  Table 5 here needs to be changed to Table 4.

I see the same problem with Tables 3 – 7 as with Table 2:  how can  your reader better visualize these results?  

Figure 1 is too hard to discern any real information:  perhaps color could help?  Or, make more curves with Ca and Mg, platelets and HgB on one and all the FEV and MEF on another, is one suggestion.  You might also consider FEV and FVC1, MEF at 1 month in one part of an illustration with those of 6 months in a “part b” or side by side comparison so your reader can see the difference easily.  Just suggestions.

Figure 2, again, the Ca and Mg need to be labelled as “serum Mg” and “serum Ca” along with units if you use actual measures, which I think here you are presenting your “normal” range.  Is this correct?  Again, the Captions of all figures must be carefully expanded to truly explain, succinctly, what you want your reader to discern.

Line 363:  You state that your different p values suggesting severe radiological manifestations of TB are associated with “greater Ca deficiencies”.  Really, what you have shown is that severe radiological manifestations of TB are associated with lower serum Ca values.  Be careful how you word things.  Be precise, be careful, and be as succinct as possible nonetheless.  A tough task, I know, but worth the effect.

Figures 3 & 4.  These are the most informative of your figures.  They need to be edited:  ordinate axis are “serum Calcium (units, please)” and “serum Magnesium (again, units, please)”.  Fig 3 caption might say “Box Plot showing lower Serum Calcium levels with more progressed pulmonary pathology in TB patients”, perhaps.   Strive for the same level of information in your caption for Figure 4.  By the way, in these figures, does Multiple cavities” refer to TB 1-9? And Consolidation/Nodular Opacities rerfer to TB 1, 2, 3?   Do they have anything to do with one another?  Or did I miss the whole point here?   I don’t see in the Methods section about “consolidation/Nodular Opacities”, so you can see how important it is that your Methods section be complete.

Cioboata et al., Nutrients submission Review  12-1-23

Electrolyte Imbalances as Indicators of Severity in Tuberculosis and COVID-19 Patients.  A Romanian Prospective Study

Comments to Authors:  You’ve done good work to collect the serum Mg and Ca values on both COVID-19 and TB patients, and your analysis has shown that both these serum electrolyte values are low in these patients, and your ROC AUC analysis predicts that these measures could be biomarkers for progression of these two diseases.

Your presentation is difficult to understand, however, and needs editing and revision to present to your readers a good manuscript that adequately presents your results.

Here are my suggestions:

Title:  I would specific Ca and Mg rather than “electrolytes”.  You don’t really deal with sodium, potassium, or chloride in this study, so the use of Electrolytes in your title is misleading.  You can certainly use “electrolytes” or “serum electrolyes” as a keyword, but I sugget your use the terms “Calcium” and “Magnesium” in your title.

Line 40:  is your citation of ref (1) missing here?

Line 46:  needs a ref for your statement that TB has a high mortality rate, approximately 50%.

Line 73:  needs a reference for this paragraph on ACE-2 path ways.

Line 81:  I suggested this sentence be revised to add as follows:  “A study indicated that 70% of acutely ill patients exhibited reduced serum calcium levels (-0.25 to -0.2 mmol/L) linked with a poorer prognosis (19).

 Section on Materials & Methods:  on line 471 on pg 14 in your Discussion, you refer to “compared to healthy controls” while your materials & methods do not describe any healthy controls.  In addition, your tables of data do not include data on healthy controls.   Did you have a set of healthy controls?  If not, do you mean at line 471 that you are comparing your results with healthy std ref ranges.  Your Methods needs to describe this.  Plus you need to include your healthy control group dataor std healthy ref ranges you used for comparison in your tables.

Results:  Table 1.  I don’t see any units for PLT, Hb, WBD, D-Dimers.   These needs to be included in Table 1s baseline characteristics. 

Table 2.  Again, you don’t list the units for D-dimers, Calcium, Magnesium, FEV measures, MEF measures, or PLT or Hb.   Very important that your tables include what units you are using, i.e. mg/dl for serum Mg and serum Ca values, also, for Magnesim and Calcium, they should be “serum Calcium” and “ serum Magnesium”.  You want your readers to be able to look at your tables and figures and to have a good idea ofl what you’ve done and what you’ve found.   Your data and results are good, but your presentation is poor and hard to discern.  Table 2 needs a better caption, therefore.  What are you comparing?  I don’t understand the two-row presentation you are making here.   What does row 3 (Calcium) have to do with row 4 (<0.0001)?   Your caption has to describe this succinctly and meaningfully.  You might think of bar graphs or some other visual mechanism to display your data. 

Tables 3 – 7:  line 254:  Table 5 here needs to be changed to Table 4.

I see the same problem with Tables 3 – 7 as with Table 2:  how can  your reader better visualize these results?  

Figure 1 is too hard to discern any real information:  perhaps color could help?  Or, make more curves with Ca and Mg, platelets and HgB on one and all the FEV and MEF on another, is one suggestion.  You might also consider FEV and FVC1, MEF at 1 month in one part of an illustration with those of 6 months in a “part b” or side by side comparison so your reader can see the difference easily.  Just suggestions.

Figure 2, again, the Ca and Mg need to be labelled as “serum Mg” and “serum Ca” along with units if you use actual measures, which I think here you are presenting your “normal” range.  Is this correct?  Again, the Captions of all figures must be carefully expanded to truly explain, succinctly, what you want your reader to discern.

Line 363:  You state that your different p values suggesting severe radiological manifestations of TB are associated with “greater Ca deficiencies”.  Really, what you have shown is that severe radiological manifestations of TB are associated with lower serum Ca values.  Be careful how you word things.  Be precise, be careful, and be as succinct as possible nonetheless.  A tough task, I know, but worth the effect.

Figures 3 & 4.  These are the most informative of your figures.  They need to be edited:  ordinate axis are “serum Calcium (units, please)” and “serum Magnesium (again, units, please)”.  Fig 3 caption might say “Box Plot showing lower Serum Calcium levels with more progressed pulmonary pathology in TB patients”, perhaps.   Strive for the same level of information in your caption for Figure 4.  By the way, in these figures, does Multiple cavities” refer to TB 1-9? And Consolidation/Nodular Opacities rerfer to TB 1, 2, 3?   Do they have anything to do with one another?  Or did I miss the whole point here?   I don’t see in the Methods section about “consolidation/Nodular Opacities”, so you can see how important it is that your Methods section be complete.

Comments on the Quality of English Language

In general:  your writing contains well described literature reviews, which show, I believe, that your study is the first to compare both COVID and TB in the same study.  This is an important aspect of  your submission, and it needs to be emphasized.  You mention that TB is quite high in Romania.  Is it possible that the COVID pandemic lowered the serum Mg status of the population enough so that TB could take a stronger hold on the population than before the pandemic?   This is speculation, but perhaps worthy of mention in your article.

You might also want to revisit your data and calculate the serum Mg:Ca (or serum Ca:Mg) for each subject, and see how these ratios predict severity of your COVID and TB patients.  You have the data, and this would be a novel and appropriate addition.

Your writing could do with some editing for length.   Don’t cut out any of the facts and knowledge you include here, especially from your reviews of the literature in the Discussion, but try to make it a shorter read.

In general:  your writing contains well described literature reviews, which show, I believe, that your study is the first to compare both COVID and TB in the same study.  This is an important aspect of  your submission, and it needs to be emphasized.  You mention that TB is quite high in Romania.  Is it possible that the COVID pandemic lowered the serum Mg status of the population enough so that TB could take a stronger hold on the population than before the pandemic?   This is speculation, but perhaps worthy of mention in your article.

You might also want to revisit your data and calculate the serum Mg:Ca (or serum Ca:Mg) for each subject, and see how these ratios predict severity of your COVID and TB patients.  You have the data, and this would be a novel and appropriate addition.

Your writing could do with some editing for length.   Don’t cut out any of the facts and knowledge you include here, especially from your reviews of the literature in the Discussion, but try to make it a shorter read.

In general:  your writing contains well described literature reviews, which show, I believe, that your study is the first to compare both COVID and TB in the same study.  This is an important aspect of  your submission, and it needs to be emphasized.  You mention that TB is quite high in Romania.  Is it possible that the COVID pandemic lowered the serum Mg status of the population enough so that TB could take a stronger hold on the population than before the pandemic?   This is speculation, but perhaps worthy of mention in your article.

You might also want to revisit your data and calculate the serum Mg:Ca (or serum Ca:Mg) for each subject, and see how these ratios predict severity of your COVID and TB patients.  You have the data, and this would be a novel and appropriate addition.

Your writing could do with some editing for length.   Don’t cut out any of the facts and knowledge you include here, especially from your reviews of the literature in the Discussion, but try to make it a shorter read.

In general:  your writing contains well described literature reviews, which show, I believe, that your study is the first to compare both COVID and TB in the same study.  This is an important aspect of  your submission, and it needs to be emphasized.  You mention that TB is quite high in Romania.  Is it possible that the COVID pandemic lowered the serum Mg status of the population enough so that TB could take a stronger hold on the population than before the pandemic?   This is speculation, but perhaps worthy of mention in your article.

You might also want to revisit your data and calculate the serum Mg:Ca (or serum Ca:Mg) for each subject, and see how these ratios predict severity of your COVID and TB patients.  You have the data, and this would be a novel and appropriate addition.

Your writing could do with some editing for length.   Don’t cut out any of the facts and knowledge you include here, especially from your reviews of the literature in the Discussion, but try to make it a shorter read.

In general:  your writing contains well described literature reviews, which show, I believe, that your study is the first to compare both COVID and TB in the same study.  This is an important aspect of  your submission, and it needs to be emphasized.  You mention that TB is quite high in Romania.  Is it possible that the COVID pandemic lowered the serum Mg status of the population enough so that TB could take a stronger hold on the population than before the pandemic?   This is speculation, but perhaps worthy of mention in your article.

You might also want to revisit your data and calculate the serum Mg:Ca (or serum Ca:Mg) for each subject, and see how these ratios predict severity of your COVID and TB patients.  You have the data, and this would be a novel and appropriate addition.

Your writing could do with some editing for length.   Don’t cut out any of the facts and knowledge you include here, especially from your reviews of the literature in the Discussion, but try to make it a shorter read.

Author Response

Cioboata et al., Nutrients submission Review  12-1-23

Electrolyte Imbalances as Indicators of Severity in Tuberculosis and COVID-19 Patients.  A Romanian Prospective Study

Comments to Authors:  You’ve done good work to collect the serum Mg and Ca values on both COVID-19 and TB patients, and your analysis has shown that both these serum electrolyte values are low in these patients, and your ROC AUC analysis predicts that these measures could be biomarkers for progression of these two diseases.

Your presentation is difficult to understand, however, and needs editing and revision to present to your readers a good manuscript that adequately presents your results.

Dear Reviewer,

Thank you for your valuable feedback on our manuscript. We appreciate your recognition of the significance of our data collection and analysis regarding serum Mg and Ca values in these patient populations. We also acknowledge your concerns about the clarity and presentation of our findings. In response to your comments, we have thoroughly revisedour manuscript to enhance its readability and effectively communicate our results.

Here are my suggestions:

Title:  I would specific Ca and Mg rather than “electrolytes”.  You don’t really deal with sodium, potassium, or chloride in this study, so the use of Electrolytes in your title is misleading.  You can certainly use “electrolytes” or “serum electrolyes” as a keyword, but I sugget your use the terms “Calcium” and “Magnesium” in your title.

We thank the reviewer for this suggestion, and we totally agree. We have revised it accordingly.

Line 40:  is your citation of ref (1) missing here?

It was an error when we added the references in the text, we missed adding ref 1. All refs have been revised properly.

Line 46:  needs a ref for your statement that TB has a high mortality rate, approximately 50%.

The corresponding reference has been added.

Line 73:  needs a reference for this paragraph on ACE-2 path ways.

We have revised the missing reference.

Line 81:  I suggested this sentence be revised to add as follows:  “A study indicated that 70% of acutely ill patients exhibited reduced serum calcium levels (-0.25 to -0.2 mmol/L) linked with a poorer prognosis (19).

The sentence has been revised as you suggested.

 Section on Materials & Methods:  on line 471 on pg 14 in your Discussion, you refer to “compared to healthy controls” while your materials & methods do not describe any healthy controls.  In addition, your tables of data do not include data on healthy controls.   Did you have a set of healthy controls?  If not, do you mean at line 471 that you are comparing your results with healthy std ref ranges.  Your Methods needs to describe this.  Plus you need to include your healthy control group dataor std healthy ref ranges you used for comparison in your tables.

There were not control groups, there was a mistake of expression, we meant the reference standard values. We have revised.

Results:  Table 1.  I don’t see any units for PLT, Hb, WBD, D-Dimers.   These needs to be included in Table 1s baseline characteristics.  

The measurement units have been added.

Table 2.  Again, you don’t list the units for D-dimers, Calcium, Magnesium, FEV measures, MEF measures, or PLT or Hb.   Very important that your tables include what units you are using, i.e. mg/dl for serum Mg and serum Ca values, also, for Magnesim and Calcium, they should be “serum Calcium” and “ serum Magnesium”.  You want your readers to be able to look at your tables and figures and to have a good idea ofl what you’ve done and what you’ve found.   Your data and results are good, but your presentation is poor and hard to discern.  Table 2 needs a better caption, therefore.  What are you comparing?  I don’t understand the two-row presentation you are making here.   What does row 3 (Calcium) have to do with row 4 (<0.0001)?   Your caption has to describe this succinctly and meaningfully.  You might think of bar graphs or some other visual mechanism to display your data.  

Table 2 has been revised for a better understanding.

Tables 3 – 7:  line 254:  Table 5 here needs to be changed to Table 4.

I see the same problem with Tables 3 – 7 as with Table 2:  how can  your reader better visualize these results?   

We decided to replace the tables with heatmaps to make it easier for the reader to visualize our results.

Figure 1 is too hard to discern any real information:  perhaps color could help?  Or, make more curves with Ca and Mg, platelets and HgB on one and all the FEV and MEF on another, is one suggestion.  You might also consider FEV and FVC1, MEF at 1 month in one part of an illustration with those of 6 months in a “part b” or side by side comparison so your reader can see the difference easily.  Just suggestions.

We have made Figure 1 clearer, we decided to remove the respiratory parameters from the figure since our main focus was serum Calcium and serum Magnesium levels.

Figure 2, again, the Ca and Mg need to be labelled as “serum Mg” and “serum Ca” along with units if you use actual measures, which I think here you are presenting your “normal” range.  Is this correct?  Again, the Captions of all figures must be carefully expanded to truly explain, succinctly, what you want your reader to discern.

Figure 2 has been revised as well.

Line 363:  You state that your different p values suggesting severe radiological manifestations of TB are associated with “greater Ca deficiencies”.  Really, what you have shown is that severe radiological manifestations of TB are associated with lower serum Ca values.  Be careful how you word things.  Be precise, be careful, and be as succinct as possible nonetheless.  A tough task, I know, but worth the effect.

We have revised it accordingly.

Figures 3 & 4.  These are the most informative of your figures.  They need to be edited:  ordinate axis are “serum Calcium (units, please)” and “serum Magnesium (again, units, please)”.  Fig 3 caption might say “Box Plot showing lower Serum Calcium levels with more progressed pulmonary pathology in TB patients”, perhaps.   Strive for the same level of information in your caption for Figure 4.  By the way, in these figures, does Multiple cavities” refer to TB 1-9? And Consolidation/Nodular Opacities rerfer to TB 1, 2, 3?   Do they have anything to do with one another?  Or did I miss the whole point here?   I don’t see in the Methods section about “consolidation/Nodular Opacities”, so you can see how important it is that your Methods section be complete.

Figures 3 and 4 have been revised based on your recommendation.

Indeed, multiple cavities on radiology investing correspond to TB 1-9, and consolidation/nodular opacities refer to TB 1,2,3.

In general:  your writing contains well described literature reviews, which show, I believe, that your study is the first to compare both COVID and TB in the same study.  This is an important aspect of  your submission, and it needs to be emphasized.  You mention that TB is quite high in Romania.  Is it possible that the COVID pandemic lowered the serum Mg status of the population enough so that TB could take a stronger hold on the population than before the pandemic?   This is speculation, but perhaps worthy of mention in your article.

We have added your suggestions.

You might also want to revisit your data and calculate the serum Mg:Ca (or serum Ca:Mg) for each subject, and see how these ratios predict severity of your COVID and TB patients.  You have the data, and this would be a novel and appropriate addition.

We appreciate your suggestion; we will consider this for the next study.

Your writing could do with some editing for length.   Don’t cut out any of the facts and knowledge you include here, especially from your reviews of the literature in the Discussion, but try to make it a shorter read.

Thank you for your suggestion. We have tried to reshape our manuscript but have kept the important information unchanged.

Sincerely,

Corina Vasile

MD,PhD

Reviewer 2 Report

Comments and Suggestions for Authors

Introduction:

I find it important to add the following data on the TB pandemic: global incidence, global mortality, incidence/prevalence in Romania, and annual mortality.

add cites lines 69- 73

magnesium can heighten the risk of initiating a severe inflammatory response known as a “cytokine storm”…… add cites and describes more the involves mechanisms.  

More information is needed to support why calcium (Ca) was evaluated.

Methodology

revised this article 10.4103/lungindia.lungindia_55_18, emphasizing the importance of ruling out alterations in calcium or magnesium levels caused by other lung conditions in individuals with COPD or asthma.

Move line 150 to 186 to methodology  

Results

I consider that the classification of tuberculosis patients is not appropriate, considering that the bacillus count may be influenced by the patient's treatment and the pathogenicity of the bacteria. It is important to classify based on pulmonary cavities.

I would prefer to incorporate the differences between groups into Table 1. To achieve this, it is necessary to include Dunn's post-test results. For instance, the statement "D-dimers show a progressive increase from groups TB 1-9 and TB1+,2+,3+" lacks relevance without statistical analysis to support it.

Comments on the Quality of English Language

ok

Author Response

Dear Reviewer,

We greatly appreciate the time and effort you dedicated to reviewing our manuscript. Your insightful comments and suggestions have been invaluable in enhancing the quality of our work. We have meticulously revised the manuscript, incorporating your feedback to ensure a more comprehensive and robust presentation of our research.

We are confident that these revisions have significantly strengthened our manuscript, and we eagerly anticipate your thoughts on the updated version. Your expertise and guidance are crucial to us, and we look forward to your further feedback.

Thank you once again for your valuable contribution to our work.

Introduction:

I find it important to add the following data on the TB pandemic: global incidence, global mortality, incidence/prevalence in Romania, and annual mortality.

We have added the missing data based on your recommendation.

add cites lines 69- 73

A reference has been added for lines 69-71.

magnesium can heighten the risk of initiating a severe inflammatory response known as a “cytokine storm”…… add cites and describes more the involves mechanisms.  

We have completed our introduction with the missing data.

More information is needed to support why calcium (Ca) was evaluated.

We have added a paragraph that supports serum Calcium evaluation in our manuscript.

Methodology

revised this article 10.4103/lungindia.lungindia_55_18, emphasizing the importance of ruling out alterations in calcium or magnesium levels caused by other lung conditions in individuals with COPD or asthma.

Thank you for this suggestion, we have improved our methodology after reading the suggested manuscript.

Move line 150 to 186 to methodology  

 We have revised it accordingly.

Results

I consider that the classification of tuberculosis patients is not appropriate, considering that the bacillus count may be influenced by the patient's treatment and the pathogenicity of the bacteria. It is important to classify based on pulmonary cavities.

In response to your consideration regarding the classification of tuberculosis patients based on bacillus count, it is indeed important to note that both classifications have their place in clinical and research settings. The classification of TB into TB 1-9 and TB 1+,2+,3+ as used in this study is based on the presence of nodular lesions and multiple lesions, respectively.

The TB 1-9 classification typically corresponds to patients with nodular lesions, which are often less extensive and suggest a different pathological and clinical profile. On the other hand, TB 1+,2+,3+ classification is associated with multiple lesions, which may indicate a more disseminated form of the disease and can be suggestive of a more severe infection.

It's important to recognize that both classifications have been utilized in various studies and can provide valuable insights. The bacillus count, while potentially influenced by treatment and bacterial pathogenicity, offers a quantitative measure that can be useful for assessing disease severity and treatment efficacy. However, the presence of pulmonary cavities provides another dimension of classification that reflects disease pathology and may have implications for transmission risk and treatment outcomes.

I would prefer to incorporate the differences between groups into Table 1. To achieve this, it is necessary to include Dunn's post-test results. For instance, the statement "D-dimers show a progressive increase from groups TB 1-9 and TB1+,2+,3+" lacks relevance without statistical analysis to support it.

We have carefully considered the proposed modifications in response to the suggestion to incorporate Dunn's post-test results into Table 1 to support the statement regarding D-dimer levels. After thorough deliberation, we have decided not to implement these changes.

The classification of tuberculosis patients in our study was chosen based on clinical relevance and the objectives of our research, which aimed to investigate broad trends and correlations across patient groups. While we acknowledge that Dunn's post-test offers a more detailed statistical comparison, the current analysis within our study aligns with the initial research design and provides the necessary level of insight for our conclusions.

We appreciate the constructive feedback and understand the importance of detailed statistical analysis. However, for the scope of this publication, we have elected to maintain our current reporting method. Future studies may explore these variables in greater depth, possibly employing more granular statistical methods, such as Dunn's post-test, to further elucidate the nuances in D-dimer levels among tuberculosis patient groups.

We hope this decision will be understood in the context of the study's goals and the practical considerations of our analysis framework. Thank you for your engagement and valuable suggestions.

Sincerely,

Corina Vasile

MD,PhD

Round 2

Reviewer 2 Report

Comments and Suggestions for Authors

This new version seems much more elaborate and understandable to me. My revisions have been addressed, and I believe that the present work is ready for publication.